# The Feature Speed Formula: a flexible approach to scale hyper-parameters of deep neural networks

**Lénaïc Chizat**
Institute of Mathematics, EPFL
Lausanne, Switzerland
lenaic.chizat@epfl.ch

**Praneeth Netrapalli**
Google DeepMind
pnetrapalli@google.com

## Abstract

Deep learning succeeds by doing hierarchical feature learning, yet tuning hyper-parameters (HP) such as initialization scales, learning rates etc., only give indirect control over this behavior. In this paper, we introduce a key notion to predict and control feature learning: the angle $\theta_\ell$ between the feature updates and the backward pass (at layer index $\ell$). We show that the magnitude of feature updates after one GD step, at any training time, can be expressed via a simple and general *feature speed formula* in terms of this angle $\theta_\ell$, the loss decay, and the magnitude of the backward pass. This angle $\theta_\ell$ is controlled by the conditioning of the layer-to-layer Jacobians and at random initialization, it is determined by the spectrum of a certain kernel, which coincides with the Neural Tangent Kernel when $\ell =$ depth. Given $\theta_\ell$, the feature speed formula provides us with rules to adjust HPs (scales and learning rates) so as to satisfy certain dynamical properties, such as feature learning and loss decay. We investigate the implications of our approach for ReLU MLPs and ResNets in the large width-then-depth limit. Relying on prior work, we show that in ReLU MLPs with iid initialization, the angle degenerates with depth as $\cos(\theta_\ell) = \Theta(1/\sqrt{\ell})$. In contrast, ResNets with branch scale $O(1/\sqrt{\text{depth}})$ maintain a non-degenerate angle $\cos(\theta_\ell) = \Theta(1)$. We use these insights to recover key properties of known HP scalings (such as $\mu$P), and also introduce a new HP scaling for large depth ReLU MLPs with favorable theoretical properties.

## 1 Introduction

The ability of deep Neural Networks (NNs) to learn hierarchical representations of their inputs has been argued to be behind their strong performance in data-intensive machine learning tasks [LeCun et al., 2015]. Yet, the process via which gradient-based training leads to feature learning remains mysterious and defies our intuition; some architectures can even reach zero loss without feature learning at all [Jacot et al., 2018]. This limited understanding makes it difficult to design NNs architectures and hyper-parameters (HP) scalings, and begs the development of tools to quantify feature learning.

In this paper, we demonstrate that the *backward-feature angle* (BFA) $\theta_\ell$ between the feature updates and the backward pass (at layer index $\ell$) is a central object in this quest. Indeed, we show that the magnitude of feature updates after one GD step, at any training time, can be expressed via a simple *feature speed formula* in terms of this angle, the loss decay and the magnitude of the backward pass. Given the knowledge of $\theta_\ell$, this leads to a general approach to quantify key dynamical properties of the training dynamics of NNs – such as the speed of feature learning and loss decay – and to characterize the HP scalings satisfying these properties.

**Contributions** Our contributions are the following:

38th Conference on Neural Information Processing Systems (NeurIPS 2024).

- We prove the *feature speed formula* (Thm 2.1) which quantifies the feature updates in terms of the BFA $\theta_\ell$, the loss decay and the magnitude of the backward pass at layer $\ell$. This formula, valid in any architecture and with an elementary proof, helps exploring the space of HP scalings, and understanding when feature learning arises.

- In Section 3, we develop tools to quantify the BFA. In particular, we show that, in the *batch-size* 1 *case*, $\theta_\ell$ can be estimated in terms of the spectrum of the backward to feature kernel (BFK) $K_\ell$ and is related to the conditioning of layer-to-layer Jacobians (Thm. 3.2). We study the case of MLPs and ResNets at random initialization, and obtain that for a depth $L$, $\cos(\theta_L) = \Theta(L^{-1/2})$ for ReLU MLPs (Prop. 5.1, exploiting a result in Jelassi et al. [2023]) and that $\cos(\theta_L) = \Theta(1)$ for linear ResNets with branch scale $O(L^{-1/2})$ (Prop. 5.2).

- In Section 4, we consider several properties of NN training dynamics that can be conveniently studied with our tools, including feature learning and loss decay. Enforcing these properties leads to explicit contraints on the magnitude of the forward, backward pass and learning rates in general architectures (Prop. 4.1).

- In Section 5, we show how various HP scalings for large width-then-depth MLPs and ResNets can be characterized by enforcing these properties. In particular we recover depth $\mu$P [Bordelon et al., 2023, Yang et al., 2023b] for ResNets (Table 2) and, for ReLU MLP, we introduce a scaling with output scale $\frac{\sqrt{\text{depth}}}{\text{width}}$ (Table 1) that does not suffer from vanishing loss decay, in contrast to the one studied in [Jelassi et al., 2023].

- Finally, in Section 6 we develop a more "axiomatic" approach: starting from a minimal list of desiderata, which include a notion of gradient stability, we show that we recover, in a certain extent, the convenient properties considered in Section 4. This section focuses on homogeneous architectures for which we show, along the way, a *backward speed formula* (Prop. 6.1) and an invariance under block-wise rescaling (Prop. 6.2).

**Related work**   The theory of NNs has recently benefited from important insights from asymptotic analyses in the large width and/or depth limits. Our work is in the continuity of those.

Analyses of wide and deep NNs at random initialization led to identifying critical initialization scalings that enable signal propagation [Poole et al., 2016, Hanin and Rolnick, 2018, Hanin, 2018]. They also identified dynamical isometry [Pennington et al., 2018], namely the concentration of the singular spectrum of the layer-to-layer Jacobians around 1, as an important indicator of training performance. Our analysis gives a concrete justification of the link between dynamical isometry and successful training, as we show that it is related to the alignment between the backward pass and feature updates. These questions have also been studied in ResNets, see e.g. [Hayou et al., 2021, Marion et al., 2022, Li et al., 2021] for signal propagation and [Tarnowski et al., 2019, Ling and Qiu, 2019] for dynamical isometry.

In 2018, two viewpoints for the dynamics of wide NNs were simultaneously introduced: a feature learning limit for two-layer MLPs [Mei et al., 2018, Chizat and Bach, 2018, Rotskoff and Vanden-Eijnden, 2018] and a limit without feature learning for general NNs [Jacot et al., 2018, Du et al., 2018, Allen-Zhu et al., 2019]. These works highlighted the crucial role of HP scalings – learning rates and initialization – in the behavior of large NNs [Chizat et al., 2019].

In order to classify HPs scalings, [Yang and Hu, 2021] formulated the *maximal update $\mu$-criterion* (it is part of the properties we study in Section 4). This criterion led to a full classification of HP scalings in the infinite hidden width limit (at fixed depth), and singled-out the so-called $\mu$-parameterization ($\mu$P) as ideal for this criterion. We note that, provided alignment holds, our analysis allows in particular to characterize $\mu$P in an elementary way. See also [Yang et al., 2023a] for another simple derivation of $\mu$P using matrix spectral norms (but that does not give tight control on the magnitude of feature learning and does not a priori apply to the large depth asymptotics). Several works have since shown the practical value of these analyses in predicting the behavior of NNs [Vyas et al., 2023] and improving HP tuning [Yang et al., 2021].

When restricted to the output layer of a NN, our notion of alignment/BFA coincides with that studied in Baratin et al. [2021], Atanasov et al. [2021], Lou et al. [2022], Wang et al. [2022] and the BFK we consider coincides with the NTK [Jacot et al., 2018]. We extend these concepts to study and quantify feature learning at *any* layer (not just at the output layer). Here we focus on the batch-size 1 setting, but the large batch-size setting of these works is a natural next direction for our analysis.

Finally, several recent works have studied feature learning in infinite width and depth NNs, starting with [Jelassi et al., 2023] for MLPs, and [Bordelon et al., 2023, Yang et al., 2023b] for ResNets. The two latter identified the $1/\sqrt{\text{depth}}$ branch scaling as providing desirable properties, in particular that of *HP transfer* [Yang et al., 2021]. These works take the infinite width limit as a first step in their analysis, before studying the resulting objects, resulting in a technical analysis. In our approach, we first take the step-size to $0$ (as in [Jelassi et al., 2023]) and study in detail the structure of the back-propagation equations, before taking the large width-then-depth limit, as a last step.

**Notations** For integers $a, b \in \mathbb{Z}$, we write $[a : b] = \{a, \ldots, b\}$. For any vector $x \in \mathbb{R}^m$ we denote by $\|x\|_{\text{rms}} := m^{-1/2}\|x\|_2$ its root mean-square (RMS) norm. We use this as a proxy for the typical entry size of a vector, which is justified as long as that vector is dense.

## 2 The Feature Speed Formula

Consider a depth-$L$ NN architecture defined by the recursion, for $\ell \in [1 : L]$,
$$f_0 \in \mathbb{R}^{m_0}, \qquad f_\ell = T_\ell(f_{\ell-1}, w_\ell) \in \mathbb{R}^{m_\ell}, \qquad \mathcal{L} = \text{loss}(f_L) \in \mathbb{R} \qquad (1)$$
where $w_\ell \in \mathbb{R}^{p_\ell}$ are trainable parameters and we assume that the maps $T_\ell : \mathbb{R}^{m_{\ell-1}} \times \mathbb{R}^{p_\ell} \to \mathbb{R}^{m_\ell}$ admit elementary (log-exp) selections[1] [Bolte and Pauwels, 2020]. By flattening the tensors, one can encode most practical NN architectures in Eq. (1). For instance, $m_0$ is typically the product of batch-size (or context length) with input dimension. We denote by $b_\ell = \left(\frac{\partial \mathcal{L}}{\partial f_\ell}\right)^\top \in \mathbb{R}^{m_\ell}$ the vectors of the backward pass. A gradient descent (GD) step with layerwise learning-rate (LR) $\eta_\ell \cdot \delta t > 0$ for $\ell \in [1 : L]$ consists in adding to each $w_\ell$ the update
$$\delta w_\ell = -\eta_\ell \cdot \delta t \cdot \nabla_\ell \mathcal{L} = -\eta_\ell \cdot \delta t \cdot \left(\frac{\partial \mathcal{L}}{\partial w_\ell}\right)^\top.$$

We are interested on the evolution of the NN over a single GD step with infinitesimally small step-size $\delta t \ll 1$. For any quantity $x$ associated to the NN, we denote $\dot{x}$ its instantaneous velocity $\dot{x} := \lim_{\delta t \downarrow 0} \frac{\delta x}{\delta t}$ when it exists. In particular, we have $\dot{w}_\ell = -\eta_\ell \nabla_\ell \mathcal{L}$.

The following identity is the seed of our approach. It expresses at any training time the speed of features in terms of other interpretable quantities, including the *backward to feature angle* (BFA) $\theta_v$.

**Theorem 2.1** (Feature speed formula). *Let $v \in [1 : L]$. If $\sum_{\ell \leq v} \eta_\ell \|\nabla_\ell \mathcal{L}\|_2^2 = 0$ then $\dot{f}_v = 0$. Otherwise, the (non-oriented) angle $\theta_v$ between $\dot{f}_v$ and $-b_v$ is well defined in $[0, \pi/2[$ and it holds*
$$\|\dot{f}_v\|_2 = \frac{\sum_{\ell \leq v} \eta_\ell \|\nabla_\ell \mathcal{L}\|_2^2}{\cos(\theta_v) \cdot \|b_v\|_2}. \qquad (2)$$

*Proof.* By the chain rule, we have $\dot{f}_v = \sum_{\ell \leq v} \frac{\partial f_v}{\partial w_\ell} \dot{w}_\ell = -\sum_{\ell \leq v} \eta_\ell \frac{\partial f_v}{\partial w_\ell}\left(\frac{\partial \mathcal{L}}{\partial w_\ell}\right)^\top$. It follows
$$-b_v^\top \dot{f}_v = \sum_{\ell \leq v} \eta_\ell \frac{\partial \mathcal{L}}{\partial f_v} \frac{\partial f_v}{\partial w_\ell}\left(\frac{\partial \mathcal{L}}{\partial w_\ell}\right)^\top = \sum_{\ell \leq v} \eta_\ell \left(\frac{\partial \mathcal{L}}{\partial w_\ell}\right)\left(\frac{\partial \mathcal{L}}{\partial w_\ell}\right)^\top = \sum_{\ell \leq v} \eta_\ell \|\nabla_\ell \mathcal{L}\|_2^2. \qquad (3)$$

Clearly, if $\sum_{\ell \leq v} \eta_\ell \|\nabla_\ell \mathcal{L}\|_2^2 = 0$ then $\dot{w}_\ell = 0$ for $\ell \leq v$ and thus $\dot{f}_v = 0$. Otherwise $\theta_v$ is well defined and it holds $\|b_v\|_2 \|\dot{f}_v\|_2 \cos(\theta_v) = -b_v^\top \dot{f}_v = \sum_{\ell \leq v} \eta_\ell \|\nabla_\ell \mathcal{L}\|_2^2$ and the claim follows. (In terms of the BFK defined below, Eq. (3) is equivalent to $b_v^\top K_v b_v = \sum_{\ell \leq v} \eta_\ell \|\nabla_\ell \mathcal{L}\|_2^2$, for $v \in [1 : L]$.) $\square$

To better appreciate the content of Thm. 2.1, let us re-express it in terms of root mean-square (RMS) norms. Let $\dot{\mathcal{L}}_{\leq v} := \sum_{\ell \leq v} \eta_\ell \|\nabla_\ell\|_2^2$ be the contribution to the loss decrease of all the parameters before $f_v$ in the forward pass, and note that $\dot{\mathcal{L}}_{\leq L} = \dot{\mathcal{L}}$. Then, the identity (2) rewrites as
$$\frac{\|\dot{f}_v\|_{\text{rms}}}{\dot{\mathcal{L}}_{\leq v}} = \frac{1}{\cos(\theta_v) \cdot m_v \cdot \|b_v\|_{\text{rms}}} =: S_v. \qquad (4)$$

---

[1]This is a technical assumption that covers virtually all functions of interest in deep learning. In particular, the maps $T_\ell$ admit *selection derivatives* that are compatible with the chain rule [Bolte and Pauwels, 2020, Prop. 4] and that coincide almost everywhere with standard derivatives [Bolte and Pauwels, 2020, Prop. 3]. To simplify our exposition, we always implicitly assume that we are at a differentiability point of the maps $T_\ell$.

Here $S_v$ can be interpreted as the *sensitivity* (and, in the terminology of [Chizat et al., 2019], $1/S_v$ as the *laziness*) of the feature $v$: it is the proportionality factor between loss decay and feature speed. This formula is valid at any training time and involves three key quantities: the scale of the backward pass $\|b_v\|_{\mathrm{rms}}$, the size of the feature $m_v$, and the BFA $\theta_v$. Let us now build tools to quantify the BFA.

## 3 Quantifying the backward-feature angles (BFA)

Information about the BFA $\theta_v$ can be gained from the Backward to Feature Kernel (BFK).

**Definition 3.1** (Backward to Feature Kernel). *For $v \in [1 : L]$, the BFK is the psd matrix defined as*

$$K_v := \sum_{\ell \leq v} \eta_\ell \Big(\frac{\partial f_v}{\partial w_\ell}\Big)\Big(\frac{\partial f_v}{\partial w_\ell}\Big)^\top \in \mathbb{R}^{m_v \times m_v}. \tag{5}$$

By construction, it holds $\dot{f}_v = -K_v b_v$. In other words, the BFK takes a backward pass vector as input and returns the (negative of the) feature velocity. For $v = L$, $K_v$ coincides with the Neural Tangent Kernel [Jacot et al., 2018]. We now show how the sprectrum of $K_v$ relates to BFA.

**Theorem 3.2.** *Let $\lambda_1 \geq \cdots \geq \lambda_{m_v} \geq 0$ be the sorted eigenvalues of $K_v$ and let $M_p := \frac{1}{m_v} \sum_{i=1}^{m_v} \lambda_i^p$ be its spectral moments. It holds $\frac{\lambda_{m_v}}{\lambda_1} \leq \cos(\theta_v) \leq 1$. Moreover, if $b_v$ is Gaussian and independent from $K_v$, then as $m_v \to \infty$,*

$$\cos(\theta_v) \xrightarrow{pr.} \frac{M_1}{\sqrt{M_2}}.$$

*as soon as $\sqrt{M_2}/M_1$ and $\sqrt{M_4}/M_2$ are uniformly bounded (i.e. are upper bounded by some $C > 0$ with probability going to $1$ as $m_v \to \infty$).*

*Proof of Thm. 3.2.* By the chain rule, it holds

$$\dot{f}_v = -\sum_{\ell \leq v} \eta_\ell \frac{\partial f_v}{\partial w_\ell} \Big(\frac{\partial \mathcal{L}}{\partial w_\ell}\Big)^\top = -\sum_{\ell \leq v} \eta_\ell \frac{\partial f_v}{\partial w_\ell} \Big(\frac{\partial f_v}{\partial w_\ell}\Big)^\top \Big(\frac{\partial \mathcal{L}}{\partial f_v}\Big)^\top,$$

hence $\dot{f}_v = -K_v b_v$. Denoting $K_v^{1/2}$ the unique psd square-root of $K_v$, it follows

$$\cos(\theta_v) = \frac{-b_v^\top \dot{f}_v}{\|\dot{f}_v\|_2 \|b_v\|_2} = \frac{\|K_v^{1/2} b_v\|_2^2}{\|K_v b_v\|_2 \|b_v\|_2}. \tag{6}$$

The first claim follows from Eq. (6) and the worst-case bounds $\|K_v b_v\|_2 \leq \lambda_1 \|b_v\|_2$ and $\|K_v^{1/2} b_v\|_2 \geq \sqrt{\lambda_{m_v}} \|b_v\|_2$. The second claim is related to the trace estimation method via random matrix-vector products [Martinsson and Tropp, 2020, Chap. 4]. We assume without loss of generality that $\mathbf{E}[\|b_v\|_2^2] = 1$ and by Lem. 3.3, we can write $Z = \|K_v^{1/2} b_v\|_2^2 = a(1 + b)$ where $a = \mathbf{E}[Z|K_v] = M_1$ and $\mathbf{E}[b^2] \to 0$ as $m_v \to \infty$. An analogous decomposition holds for $\|K_v(b_v)\|_2^2$ with $a = M_2$ and the second claim follows. $\qquad\square$

**Lemma 3.3.** *Let $K \in \mathbb{R}^{m \times m}$ be a (potentially random) psd matrix and $a \sim \mathcal{N}(0, \frac{1}{m} I_m)$ be independent. Then $\mathbf{E}[\|Ka\|_2^2 \mid K] = M_2(K)$ and $\mathrm{Var}[\|Ka\|_2^2 \mid K] = \frac{2}{m} M_4(K)$ where $M_p(K) := \frac{1}{m} \sum_{i=1}^m \lambda_i^p$ and $\lambda_1, \ldots, \lambda_m \geq 0$ are the eigenvalues of $K$.*

The second claim expresses the BFA in terms of the spread of the spectrum of the BFK, in an asymptotically exact way. Its assumptions hold at random initialization in the large width limit of typical NNs, provided $f_v$ is directly followed by a weight-matrix multiplication in the forward pass, so that $b_v$ is the output of a random matrix/vector multiplication. Asymptotic independence can be guaranteed in quite general contexts, see Yang [2020]. For MLP or ResNets with batch-size one, we show in Section 5 that $\cos(\theta_v)$ is tightly related to the conditioning of layer-to-layer Jacobians, studied in the *dynamical isometry* literature [Pennington et al., 2017].

# 4 Ensuring feature learning in scaled NNs

## 4.1 Properties for scaled NNs

Consider a sequence of NNs and parameters as in (1) with some diverging architectural parameters such as depth or width. We refer to such a sequence as a *scaled NN*. In search of the optimal scaling of NNs, it is crucial to understand how HP scalings influence the properties of the training dynamics. In this section, we discuss the following properties:

(SP) **Signal propagation.** It holds $\|f_v\|_{\mathrm{rms}} = \Theta(1)$ for $v \in [1 : L-1]$.

(FL) **Feature learning.** It holds $\|\dot{f}_{L-1}\|_{\mathrm{rms}} = \Theta(1)$.

(LD) **Loss decay.** It holds $-\dot{\mathcal{L}} = \Theta(1)$.

(BC) **Balanced contributions.** It holds $\eta_\ell \|\nabla_\ell \mathcal{L}\|_2^2 = \Theta(\eta_{\ell'} \|\nabla_{\ell'} \mathcal{L}\|_2^2)$ for any $\ell, \ell' \in [1 : L]$.

We discuss these specific properties because they are amenable to our tools and enforcing them requires $(L-1) + 1 + 1 + (L-1) = 2L$ degrees of freedom, which exactly matches the number of free HPs if one counts one scale HP (such as the variance of the weights) and one LR per block. One may wonder if property (BC) is truly desirable: this is the topic of Section 6, where we adopt a more axiomatic approach and recover, for homogeneous architectures, (a more general version of) property (BC) as a consequence of enforcing *gradient stability*. Also, while enforcing these properties is reasonable when increasing depth and width, they should be rethought for other asymptotics.

Property (SP) specifies $L-1$ scale HPs, but leaves the scale of $f_L$ free. The reason for not including $f_L$ in (SP) is that $\|f_L\|_{\mathrm{rms}} = o(1)$ does not lead to vanishing gradient in general, so this behavior should not be excluded a priori. How should one then fix the scale of the output? The next proposition shows that for property (FL) to hold, the quantity that should be suitably normalized is the norm of the backward pass.

**Proposition 4.1.** *A scaled NN* (1) *satisfies (FL), (LD), and (BC) if and only if*

$$\|b_{L-1}\|_{\mathrm{rms}} = \Theta\Big(\frac{1}{m_{L-1} \cdot \cos(\theta_{L-1})}\Big) \tag{7}$$

*and*

$$\forall \ell \in [1 : L], \ \eta_\ell = \Theta\Big(\frac{1}{L\|\nabla_\ell \mathcal{L}\|_2^2}\Big). \tag{8}$$

*Proof.* Property (LD) requires $\sum_{\ell=1}^{L} \eta_\ell \|\nabla_\ell \mathcal{L}\|_2^2 = -\dot{\mathcal{L}} = \Theta(1)$ and (BC) requires the terms in the sum to be balanced, this leads to Eq. (8). Now by Thm. 2.1, property (FL) requires

$$\|\dot{f}_{L-1}\|_{\mathrm{rms}} = \frac{\sum_{\ell=1}^{L-1} \eta_\ell \|\nabla_\ell \mathcal{L}\|_2^2}{\cos(\theta_{L-1}) \cdot m_{L-1} \cdot \|b_{L-1}\|_{\mathrm{rms}}} = \Theta(1) \tag{9}$$

which leads to (7). Conversely, it is clear that Eq. (8) and (7) imply (FL), (LD) and (BC). $\square$

## 4.2 Towards automatic HP scaling

The criterion of Prop. (4.1), complemented with the property (SP), suggest a method to automatically adjust the scales and learning rates in any architecture. In general, properties (SP), (FL), (BC) and (LD) can be enforced as follows:

- **(SP): Forward layer normalization.** Enforcing property (SP) can be done along with the computation of the forward pass, this is the usual layer normalization.

- **(FL): Backward layer normalization.** Provided $\theta_{L-1}$ is known or measured, Eq. (7) can be enforced via a *backward* analog to layer normalization: one inserts a scaling factor in the forward pass between $f_{L-1}$ and $f_L$, adjusted so that Eq. (7) holds.

- **(BC) & (LD): Scale invariant learning rates.** Directly tune the LRs via Eq. (8).

We refer to the resulting scaling as FSC as it normalizes the **F**orward pass, the **S**ensitivities and the **C**ontributions. Let us make some observations regarding the scale invariant LRs:

- **Link with Polyak step-size.** In convex optimization, to minimize a convex and Lipschitz continuous function $f : \mathbb{R}^d \to \mathbb{R}$ such that $\min_{x \in \mathbb{R}^d} f(x) = 0$, the Polyak-step-size [Polyak, 1987, Hazan and Kakade, 2019] for the GD algorithm $x_{t+1} = x_t - \eta_t \nabla f(x_t)$ is given by $\eta_t = \frac{f(x_t)}{\|\nabla f(x_t)\|_2^2}$. With this step-size, GD achieves the optimal convergence rate for first order methods over the class of convex and Lipschitz functions. Eq. (8) require a layerwise version of this step-size schedule.

- **Interplay with adaptive methods (Adagrad [Duchi et al., 2011], ADAM [Kingma and Ba, 2015]).** Adaptive gradient method typically divide the gradient by a quantity which grows *linearly* rather than quadratically with the norm of the gradient, such as $\delta W_\ell = -\tilde{\eta}_\ell \cdot \frac{\nabla_\ell \mathcal{L}}{\|\nabla_\ell \mathcal{L}\|_{\mathrm{rms}}}$. For such an algorithm, properties (BC) and (LD), suggests the LR $\tilde{\eta}_\ell = \Theta(\frac{\|\nabla_\ell \mathcal{L}\|_{\mathrm{rms}}}{L \cdot \|\nabla_\ell \mathcal{L}\|_2^2})$, in place of Eq. (8).

- **Scale invariance and $-2$ homogeneity.** These LRs arise naturally when one wants to make the gradient descent invariant to how scale is enforced (via initialization scale or via scaling factors). We show in App. C that any choice of LR that leads to this invariance must be a positively homogeneous function of the layer-wise gradient of degree $-2$, as in Eq. (8). We also show in Prop. 6.2 that these LRs make homogeneous architectures invariant to the choice of layer-wise scalings $\sigma_1, \ldots, \sigma_L$, given a fixed global scale $\prod_{\ell=1}^L \sigma_\ell$.

## 5 Scaling width and depth of MLPs and ResNets

### 5.1 BFA for single input MLPs and ResNets at initialization

**Multilayer Perceptron** Consider a ReLU MLP architecture with a single input $x = g_0 \in \mathbb{R}^d$ and a forward pass given, for $\ell \in [1 : L - 1]$, by

$$f_\ell = W_\ell g_{\ell-1}, \qquad g_\ell = \phi(f_\ell), \qquad f_L = W_L g_{L-1}, \qquad \mathcal{L} = \mathrm{loss}(f_L) \qquad (10)$$

where $\phi(u) = \max\{0, u\}$ is the ReLU nonlinearity and acts entrywise on vectors. The architecture HPs are the input width $m_0 = d$, the widths of the hidden layers $m_1 = \cdots = m_{L-1} = m$ (assumed equal), the output width $m_L = k$. The trainable parameters are $\forall \ell \in [1 : L]$, $W_\ell \in \mathbb{R}^{m_\ell \times m_{\ell-1}}$. Such NNs are of the form (1) and are thus covered by Thm. 2.1. Let us study their properties at random initialization under the following assumptions:

(H1) the weights $W_\ell$ are independent $\mathcal{N}(0, \sigma_\ell^2)$ random variables for $\ell \in [1 : L]$.

(H2) either $k = \Theta(1)$ or the loss is linear.

In this setting, the following statements gather consequences of results from the literature on random NNs and of Thm. 3.2 to derive the scale of the forward and backward passes and the BFA. We require (H2) as a technical assumption to avoid dealing with cases where $b_L$ strongly depends on the forward pass, where different scalings may arise[2].

In what follows we write $A = \Theta(B)$ when there exists $c, C > 0$ independent of $d, m, k, L, \|x\|_2$ and $\|b_L\|_2$ such that the probability that $A/B \in [c, C]$ goes to 1 in the specified asymptotic. The new result in the following proposition is the BFA estimate, which relies crucially on a delicate computation due to [Jelassi et al., 2023].

**Proposition 5.1** (Large width and depth MLP). *Assume (H1-2) and for $\ell \in [1 : L - 1]$, let $\sigma_\ell = \sqrt{2/m_{\ell-1}}$. As $m \to \infty$, it holds*

$$\|f_v\|_{\mathrm{rms}} = \Theta(\|x\|_{\mathrm{rms}}), \qquad \|b_v\|_2 = \Theta(\sqrt{m}\,\sigma_L\,\|b_L\|_2). \qquad (11)$$

*Moreover, if (BC) holds then $\cos(\theta_v) = \Theta(v^{-1/2})$.*

**ResNets** Consider now a ResNet with a *branch scale* parameter $\beta \in [0, 1]$, as in Li et al. [2021]: with a single input $x = f_0 \in \mathbb{R}^d$, the forward pass is given, for $\ell \in [2 : L - 1]$, by

$$f_1 = W_1 x, \quad f_\ell = \sqrt{1 - \beta^2} f_{\ell-1} + \beta W_\ell \phi(f_{\ell-1}), \quad f_L = W_L f_{L-1}, \quad \mathcal{L} = \mathrm{loss}(f_L) \qquad (12)$$

---

[2]Say, if $\mathrm{loss}(f) = \frac{1}{2}\|f\|_2^2$, we have $\|b_{L-1}\|_{\mathrm{rms}} = \|W_L^\top W_L f_{L-1}\|_{\mathrm{rms}} = \Theta(\sigma_L \max\{1, \sqrt{k/m}\}\|b_L\|_2)$ (by Lem. 3.3 and properties of the Marcenko-Pastur law), while under (H2) we have $\|b_{L-1}\|_{\mathrm{rms}} = \Theta(\sigma_L \|b_L\|_2)$.

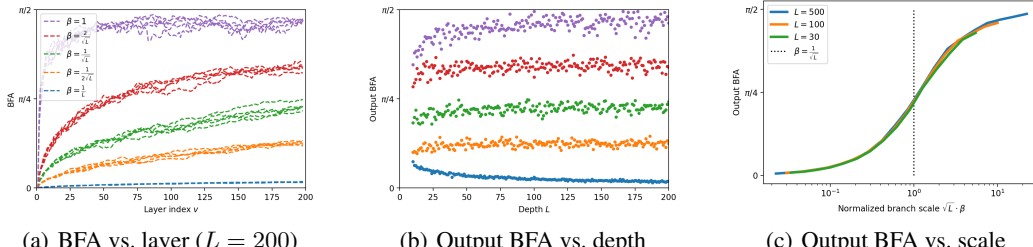

|  (a) BFA vs. layer ($L = 200$)  |  (b) Output BFA vs. depth  |  (c) Output BFA vs. scale  |

Figure 1: Backward-Feature Angle (BFA) $\theta_v$ observed at initialization in MLPs ($\beta = 1$) and ResNets (width $m = 200$), for a few random realizations. (a) for all architectures, BFA $\theta_v$ varies in the first few layers and then stabilizes. (b) BFA at output layer $\theta_{L-1}$ is asymptotically independent of depth, with a non-trivial angle only when $\beta \propto 1/\sqrt{L}$ (same color scheme as (a)). (c) for a branch scale $\beta = c/\sqrt{L}$, the factor $c$ directly determines asymptotic output BFA $\theta_{L-1}$ (averaged over 5 draws).

where $\phi(u) = u$ in our theoretical results. The architecture HPs are the input width $m_0 = d$, the widths of the hidden layers $m_1 = \cdots = m_{L-1} = m$, the output width $m_L = k$. The trainable parameters are $\forall \ell \in [1 : L]$, $W_\ell \in \mathbb{R}^{m_\ell \times m_{\ell-1}}$. When $\beta = 1$, we recover a MLP.

Here we limit ourselves to the case of linear activation where we can directly apply a result from [Marion and Chizat, 2024] to estimate the BFA. We believe that the same result and proof technique extend to the ReLU activation and other variants of ResNets; these extensions are left to future work.

**Proposition 5.2** (Large width and depth linear ResNet). *Assume (H1-2), let $\phi(x) = x$, $\beta = O(1/\sqrt{L})$ and for $\ell \in [1 : L - 1]$, let $\sigma_\ell = \Theta(1/\sqrt{m_{\ell-1}})$. As $m \to \infty$ it holds:*

$$\|f_\ell\|_{\mathrm{rms}} = \Theta(\|x\|_{\mathrm{rms}}), \qquad\qquad \|b_\ell\|_2 = \Theta\left(\sqrt{m}\sigma_L\|b_L\|_2\right). \qquad (13)$$

*Moreover, if (BC) holds then $\cos(\theta_v) = \Theta(1)$.*

**Numerical experiments**  We consider[3] one GD step in the model (12) with ReLU nonlinearity, without training $W_1$ (input dimension $d = 10$, output dimension $k = 1$, master LR $\delta t = 0.001$). Fig. 1, represent BFA, computed via $\theta_v \approx \arccos(-b_v^\top \delta f_v)$ where $\delta f_v$ is the change of feature $f_v$ after one GD step. The results are consistent with Prop. 5.1 and 5.2. Interestingly, the last plot suggests that there exists a function $\varphi : \mathbb{R}_+ \to \,]0, \pi/2[$ such that for a branch scale $\beta = c/\sqrt{L}$, the BFA converges to $\varphi(c)$ (it can be conjectured numerically that $\cos(\varphi(c)) \approx c^{-1/2}$ for $c \gg 1$).

### 5.2 Characterizing HP scalings for MLPs

Let us now discuss specific choices of HP scalings for single-input MLP architectures as in Eq. (10) (or Eq. (12) with $\beta = 1$) and at initialization. We consider 6 HPs: the scale of initialization $\sigma_1$ and LR $\eta_1$ of the input layer, the scale $\sigma_{\mathrm{hid}} := \sigma_2 = \cdots = \sigma_{L-1}$ and LRs $\eta_{\mathrm{hid}} := \eta_2 = \cdots = \eta_{L-1}$ of the hidden layers, and the scale $\sigma_L$ and LR $\eta_L$ of the output layer. The HP scalings mentioned in the next theorem are the following (see Table 1):

- **NTK**: the standard scaling with LRs adjusted to satisfy (LD) and (BC) [Jacot et al., 2018];
- **MF+$\mu$P**: the scaling proposed in [Jelassi et al., 2023] constructed by imposing the so-called "mean-field" output scale $\sigma_L \propto 1/m$ and then enforcing (FL) by adjusting the learning rates;
- **FSC**: the HP scaling singled-out by Prop. 5.3, obtained by adjusting the **F**orward scales, **S**ensitivities, and **C**ontributions.

The properties of HP scalings depend on $\|x\|_2$ and $\|b_L\|_2$. We consider two typical settings:

- (Dense) Where $\|x\|_2 = \sqrt{d}$ and $\|b_L\|_2 = \frac{1}{\sqrt{k}}$. This is representative of a dense whitened input and a RMS loss $\mathrm{loss}(f_L) = \|f_L - y\|_2^2/k$ for some dense signal $y \in \mathbb{R}^k$ with $\|y\|_{\mathrm{rms}} = \Theta(1)$ as, e.g., in image generation applications.

---

[3]Link to the Julia code to reproduce the experiments: `https://github.com/lchizat/2023-BAFU`

Table 1: HP scalings for MLPs under the *dense* setting (for the *sparse* setting, replace $k$ and $d$ by 1). For $L$ fixed, both **MF+$\mu$P** and **FSC** coincide with $\mu$P. Values in red are exact, the others are up to a multiplicative factor in $\Theta(1)$.

| | | Input | Hidden | Output |
|---|---|---|---|---|
| **FSC** | init. std. $\sigma_\ell$ | $1/\sqrt{d}$ | $\sqrt{2/m}$ | $\sqrt{kL}/m$ |
| | LR $\eta_\ell$ | $m/L^2 d$ | $1/L^2$ | $k/Lm$ |
| **MF+$\mu$P** | init. std. $\sigma_\ell$ | $1/\sqrt{d}$ | $\sqrt{2/m}$ | $\sqrt{k}/m$ |
| | LR $\eta_\ell$ | $m/L^{3/2} d$ | $1/L^{3/2}$ | $k/L^{3/2}m$ |
| **NTK** | init. std. $\sigma_\ell$ | $1/\sqrt{d}$ | $\sqrt{2/m}$ | $1/\sqrt{m}$ |
| | LR $\eta_\ell$ | $1/Ld$ | $1/Lm$ | $k/Lm$ |

Table 2: **FSC** scalings identified in Prop. 5.4 for ResNets. All HPs are specified up to a multiplicative factor in $\Theta(1)$. When $\beta = \Theta(L^{-1/2})$ and $k = d = \Theta(1)$, these scalings coincide with the so-called "depth $\mu$P" introduced in [Bordelon et al., 2023] and also studied in Yang et al. [2023b].

| | Input | Hidden | Output |
|---|---|---|---|
| init. std. $\sigma_\ell$ | $1/\sqrt{d}$ | $1/\sqrt{m}$ | $\sqrt{k}/m$ |
| LR $\eta_\ell$ | $m/Ld$ | $1/\beta^2 L$ | $k/Lm$ |

- (Sparse) Where $\|x\|_2 = 1$ and $\|b_L\|_2 = 1$. This is representative of a one-hot encoding input and the multiclass logistic loss (aka cross-entropy where $\|b_L\|_2 = \Theta(\log(k))$). This setting is typical of natural language processing tasks.

The scalings are reported in Table 1. We have also introduced scalings in terms of output width $k$ for **NTK** and **MF + $\mu$P** to ensure a non-degenerate behavior as $k \gg 1$, although these are generally not written in the literature.

**Proposition 5.3** (MLP scalings). *Under the assumptions of Prop. 5.1, the following hold at random initialization:*

  (i) *The scaling **MF+$\mu$P** satisfies (SP), (BC) , (FL) but not (LD);*

  (ii) *The scaling **NTK** satisfies (SP), (BC), (LD) but not (FL);*

  (iii) *Properties (SP), (BC), (LD), (FL) hold if and only if the scaling is **FSC**.*

This theorem identifies the new HP scaling **FSC** for deep ReLU MLP where the scale of the output layer depends on the depth. We compare empirically the sensitivities (Eq. (4)) of the various scalings in Fig. 2, and the results are consistent with theory. Finally, let us mention that even though **FSC** fixes some degeneracies of deep MLPs, other problems arise when considering multiple inputs, such as degeneracy of the conjugate kernel and NTK [Hayou et al.], which make ReLU MLPs a fundamentally flawed model at large depth. Hence, the analysis of large depth scalings for MLPs is mostly of theoretical interest.

### 5.3   Characterizing HP scalings for ResNets

We now discuss HP scalings for single-input ResNets (Eq. (12)) with $\beta = O(1/\sqrt{L})$.

**Proposition 5.4** (ResNets scalings). *Take $\beta = O(1/\sqrt{L})$, consider the same 6 degrees of freedom as in the previous section and assume that the conclusions of Prop. 5.2 holds. Then properties (SP), (BC), (LD) and (FL) hold at initialization if and only if the scalings are those given in Table 2.*

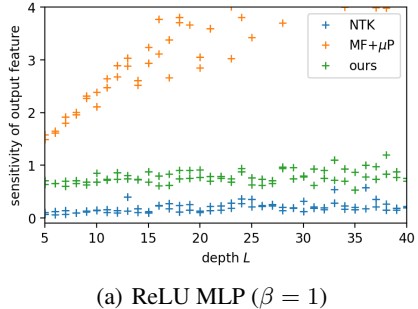
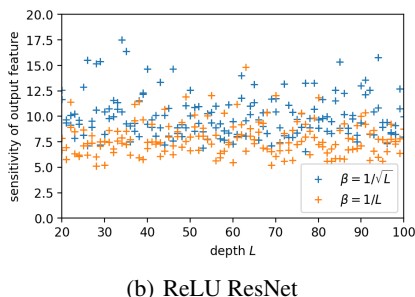

(a) ReLU MLP ($\beta = 1$)  (b) ReLU ResNet

Figure 2: Sensitivities $S_{L-1}$ (see Eq. (4)) of the last layer of activations ($g_{L-1}$ in the MLP and $f_{L-1}$ in the ResNet) computed via the formula $\|\delta f_{L-1}\|_{\mathrm{rms}}/|\delta\mathcal{L}|$ where $\delta$ denotes the change after one GD step (master learning rate $\delta t = 0.01$, $d = 10$, $n = 1$ input sample on the unit sphere and $k = 1$). (a) ReLU MLP of width $m = 400$. From our theory we have for **NTK**, $S = \Theta(1/\sqrt{m})$ (close to 0 and constant with depth); for **MF+$\mu$P** $S = \Theta(\sqrt{L})$ and for the **FSC** $S = \Theta(1)$ (b) ReLU ResNet of width $m = 400$: for both choices of branch scale, the sensitivities is stable around a nonzero value.

## 6 Minimal desiderata and stability under homogeneity

It is not clear a priori why the property (BC) studied in Section 4 should be enforced. In this section, we consider homogeneous architectures, such as ReLU MLPs, and show that a slightly more general version of (BC) is a related to a notion of *gradient stability*.

### 6.1 Stability and backward speed formula

For a general architecture of the form Eq. (1), consider the following stability property (S), which is necessary if one wants to have comparable behavior between the first GD step and the next. It is related to the usual notion of smoothness in optimization:

(S) **Stability**. It holds $\|\frac{\mathrm{d}}{\mathrm{d}t}\nabla_\ell\mathcal{L}\|_2/\|\nabla_\ell\mathcal{L}\|_2 = O(1)$ for $\ell \in [1:L]$.

We will study this property in a ReLU MLP with a single input as in Eq. (10) (the extension to linear ResNets is simple as only the BFAs change). In this case we have $\nabla_\ell\mathcal{L} = b_\ell g_{\ell-1}^\top$ and thus $\|\nabla_\ell\mathcal{L}\|_F = \|b_\ell\|_2\|g_{\ell-1}\|_2$. It follows

$$\frac{\|\frac{\mathrm{d}}{\mathrm{d}t}\nabla_\ell\mathcal{L}\|_F}{\|\nabla_\ell\mathcal{L}\|_F} \leq \frac{\|\dot{b}_\ell\|_2\|g_{\ell-1}\|_2 + \|b_\ell\|_2\|\dot{g}_{\ell-1}\|_2}{\|b_\ell\|_2\|g_{\ell-1}\|_2} = \frac{\|\dot{b}_\ell\|_2}{\|b_\ell\|_2} + \frac{\|\dot{g}_{\ell-1}\|_2}{\|g_{\ell-1}\|_2}.$$

We can thus ensure the gradient stability by ensuring, for all $\ell \in [1:L]$,

(FS) **Forward stability**. It holds $\frac{\|\dot{g}_{\ell-1}\|_2}{\|g_{\ell-1}\|_2} = O(1)$ for $\ell \in [1:L]$, and

(BS) **Backward stability**. It holds $\frac{\|\dot{b}_\ell\|_2}{\|b_\ell\|_2} = O(1)$ for $\ell \in [1:L]$.

We focus on these simpler desiderata (FS) and (BS) instead of (S) for the rest of the discussion. To estimate $\dot{b}_v$, we rely on a "backward" version of the feature speed formula that holds in 1-homogeneous NNs.

**Proposition 6.1** (Backward speed formula). *Consider a general architecture of the form* (1)*, take* $v \in [1:L]$ *and assume that the map* $f_v \to f_L$ *is positively 1-homogeneous[4]. If* $-f_L^\top\nabla^2\mathrm{loss}[f_L]\dot{f}_L + \sum_{\ell>v}\eta_\ell\|\nabla_\ell\mathcal{L}\|_2^2 = 0$ *then* $\dot{b}_v = 0$*. Otherwise, the (non-oriented) angle* $\tilde{\theta}_v$ *between* $f_v$ *and* $\dot{b}_v$ *is well defined in* $[0, \pi/2[$ *and it holds*

$$\|\dot{b}_v\|_2 = \frac{-f_L^\top\nabla^2\mathrm{loss}[f_L]\dot{f}_L + \sum_{\ell>v}\eta_\ell\|\nabla_\ell\mathcal{L}\|_2^2}{\|f_v\|_2\cos(\tilde{\theta}_v)}.$$

---

[4]For Euler's identity to hold, we also assume that its selection [Bolte and Pauwels, 2020] is 0-homogeneous.

In the context of ReLU MLPs with a linear loss, we have by differentiating the back-propagation recursion and noticing that all terms involving $\phi''$ are zero almost surely[5] that:

$$\dot{b}_v = \sum_{\ell > v} \eta_\ell \left(\frac{\partial g_{\ell-1}}{\partial f_v}\right)^\top g_{\ell-1} b_\ell^\top b_\ell = \sum_{\ell > v} \eta_\ell \|b_\ell\|_2^2 \left(\frac{\partial g_{\ell-1}}{\partial f_v}\right)^\top \left(\frac{\partial g_{\ell-1}}{\partial f_v}\right) f_v = \tilde{K}_v f_v \qquad (14)$$

where the last expression defines $\tilde{K}_v$. Reasoning as in Thm. (3.2), since $f_v$ is Gaussian at initialization and noticing that $\tilde{K}_v$ has a structure similar to that of $K_{L-v}$, we have that $\cos(\tilde{\theta}_v) = \Theta(\sqrt{L-v})$, see the details in Lem. A.1. We can thus estimate $\dot{b}_v$ just as well as $\dot{f}_v$.

### 6.2 Scale invariance for homogeneous architectures

Homogeneous architectures such as ReLU MLP satisfy scale invariance properties that are important to take into account in our discussion. The following result presents a general invariance under blockwise rescaling, provided one uses scale-invariant LRs. This is related to known invariance results under global rescaling for scale invariant losses [Van Laarhoven, 2017, Li et al., 2022, Wan et al., 2020].

**Proposition 6.2** (Invariance under block-wise rescaling)**.** *Consider a function $f_L(w_1, \ldots, w_L)$ (the NN, in our context) which is separately positively $1$-homogeneous in each of its blocks of parameters $w_\ell \in \mathbb{R}^{p_\ell}$. Let $\theta_0 = (w_1(0), \ldots, w_L(0))$ and let $\tilde{\theta}_0 = \sigma \odot \theta_0 := (\sigma_1 w_1(0), \ldots, \sigma_L w_L(0))$ for some scale factors $\sigma \in \mathbb{R}_+^L$. Let $\theta(t)$ and $\tilde{\theta}(t)$ be the iterates of GD on $\mathcal{L} : \theta \mapsto \mathrm{loss}(f_L(\theta))$ with LR satisfying $\eta_\ell(t)\|\nabla_\ell \mathcal{L}(\theta(t))\|_2^2 = \tilde{\eta}_\ell(t)\|\nabla_\ell \mathcal{L}(\tilde{\theta}(t))\|_2^2$ and starting from $\theta_0$ and $\tilde{\theta}_0$ respectively. If $\prod_{\ell=1}^L \sigma_\ell = 1$ then $\tilde{\theta}(t) = \sigma \odot \theta(t)$ for all $t \geq 1$.*

### 6.3 Characterization of admissible scalings for ReLU MLPs

In view of Prop. 6.2, for homogeneous architectures, one can ignore (SP) since any GD dynamics is equivalent to a dynamics where (SP) holds at initialization. However, if the scale of the forward pass is not normalized, (FL) needs now to be adapted to a scale-free version, as follows:

(RFL) **Relative feature learning**. It holds $\|\dot{f}_{L-1}\|_2 / \|f_{L-1}\|_2 = \Theta(1)$.

We are finally in position to gather all these insights and characterize all *admissible* scalings for ReLU MLPs, i.e. scalings that satisfy the minimal desiderata (RFL), (LD), (FS) and (BS) at initialization.

**Theorem 6.3** (Minimal desiderata for MLPs)**.** *Consider a ReLU MLP with $6$ degrees of freedom: three initialization scales $\sigma_1, \sigma_{\mathrm{hid}}, \sigma_L$ and three LRs $\eta_1, \eta_{\mathrm{hid}}, \eta_L$. Assume $\|b_L\|_2 = \|g_0\|_{\mathrm{rms}} = 1$ and a linear loss for simplicity. Then the minimal desiderata (RFL), (LD), (FS) and (BS) hold at initialization in the limit $m \to \infty$ then $L \to \infty$ if and only if*

$$(\sqrt{d}\sigma_1) \cdot (\sqrt{m/2}\sigma_{\mathrm{hid}})^{L-2} \cdot \sigma_L = \Theta(\sqrt{L}/m), \quad C_1 + C_{\mathrm{hid}} = \Theta(1) \quad \textit{and} \quad C_L = O(1),$$

*where $C_1 = \eta_1\|\nabla_1\mathcal{L}\|_2^2$, $C_{\mathrm{hid}} = \sum_{\ell=2}^{L-1} \eta_{\mathrm{hid}}\|\nabla_\ell\mathcal{L}\|_2^2$ and $C_L = \eta_L\|\nabla_L\mathcal{L}\|_2^2$. In particular, the scaling **FSC (Table 1)** satisfies these desiderata.*

## 7 Conclusion

Starting from the feature speed formula, our approach allows to conveniently recover and characterize in an elementary fashion certain properties of existing HP scalings and to discover new ones, with essentially all the technical difficulty contained in the estimation of the BFA. The limitations of our approach are related to the blind spots of Thm. (2.1): it can only quantify feature speed for (S)GD (and does not apply to variants in its current form) and at "cut nodes" in the NN architecture, where all the signal goes through (in particular, it does not apply inside the blocks of a ResNet).

In future works, besides removing these limitations, it would be interesting to have a better understanding of the BFA, both from a quantitative and a qualitative viewpoint.

---

[5] A downside of this computation is that, because of its local nature, it ignores the contributions of the Jacobian's discontinuities to $\dot{b}_L$, while they do have a "macroscopic" effect with a non-vanishing step-size. For instance, taking first the infinite width and then the small step-size limit, would give a different expression.

## Acknowledgments

This work was supported in part by the International Centre for Theoretical Sciences (ICTS) for participating in the meeting - Data Science: Probabilistic and Optimization Methods (code:ICTS/dspom2023/7)

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

# A Proofs omitted from the main text

*Proof of Lem. 3.3.* Writing $K = VDV^\top$ with $D = \mathrm{diag}(\lambda_1, \ldots, \lambda_m)$ and $V \in \mathbb{R}^{m \times m}$ orthonormal, we have $\|Ka\|_2^2 = a^\top V D^2 V^\top a$. Conditioned on $K$, the vector $u = V^\top a$ is isotropic Gaussian so $\mathbf{E}u_i^2 = \frac{1}{m}$ for $i \in [1:m]$. Hence, on the one hand

$$\mathbf{E}[\|Ka\|_2^2 \mid K] = \mathbf{E}\Big[ \sum_{i=1}^m \lambda_i^2 u_i^2 \mid (\lambda_i)_{i=1}^m \Big] = \sum_{i=1}^m \lambda_i^2 \mathbf{E}[u_i^2] = \frac{1}{m} \sum_{i=1}^m \lambda_i^2.$$

On the other hand, using the fact that the variance of a chi-square random variable is 2,

$$\mathrm{Var}[\|Ka\|_2^2 \mid K] = \mathbf{E}\Big[ \Big( \sum_{i=1}^m \lambda_i^2 (u_i^2 - 1/m) \Big)^2 \mid (\lambda_i)_{i=1}^m \Big]$$

$$= \sum_{i=1}^m \lambda_i^4 \mathbf{E}[(u_i^2 - 1/m)^2] + \sum_{i \neq j}^m \lambda_i^2 \lambda_j^2 \mathbf{E}[(u_i^2 - 1/m)(u_j^2 - 1/m)] = \frac{2}{m^2} \sum_{i=1}^m \lambda_i^4. \square$$

*Proof of Prop.5.1.* When $\beta = 1$, Eq. (11) is classical from the signal propagation literature [Poole et al., 2016, Hanin and Rolnick, 2018, Hanin, 2018] (the fluctuations around the limit have also been studied in Hanin and Nica [2020]). Note that these results are proved with $k = \Theta(1)$, but Hanin [2018] allows to conclude as well when $k$ diverges at least if the initial gradient $b_L = \left( \frac{\partial \mathcal{L}}{\partial f_L} \right)^\top \in \mathbb{R}^k$ is independent of the randomness of the weights, which is what (H2) guarantees. We note that analogous results have been derived for a variety of activation functions, and we focus on ReLU only for conciseness.

Let us now discuss the BFAs, assuming for simplicity that $\eta_1 = 0$ as the contribution of $w_1$ to the BFK is asymptotically negligible assuming (BC). We consider the BFA at $g_v$ and denote $z_v := (\partial \mathcal{L}/\partial g_v)^\top$. The main result of [Jelassi et al., 2023] can be restated as follows: with $k = d = \Theta(1)$, the choice $\sigma_L = \frac{1}{m}$ and learning-rates $\eta_\ell = \Theta(L^{-3/2})$, it holds $\|\dot{g}_{L-1}\|_{\mathrm{rms}} = \Theta(1)$. In view of (11), it holds in their setting for $\ell \in [2:L-1]$

$$\|\nabla_\ell \mathcal{L}\|_2^2 = \|b_\ell g_{\ell-1}^\top\|_2^2 = \|g_{\ell-1}\|_2^2 \|b_\ell\|_2^2 = \Theta(m_{\ell-1} \cdot m_{L-1} \cdot \sigma_L^2) = \Theta(m_{\ell-1}/m_{L-1}).$$

Using $\eta_\ell = \Theta(L^{-3/2})$, it follows $\sum_{\ell=2}^{L-1} \eta_\ell \|\nabla_\ell \mathcal{L}\|_2^2 = \Theta(L^{-1/2})$. By Thm. 2.1 and using $m_{L-1}\|z_{L-1}\|_{\mathrm{rms}} = \Theta(1)$, we get

$$\|\dot{g}_{L-1}\|_{\mathrm{rms}} = \frac{\sum_{\ell \leq L} \eta_\ell \|\nabla_\ell \mathcal{L}\|_2^2}{\cos(\theta_{L-1}) \cdot m_{L-1} \cdot \|z_{L-1}\|_{\mathrm{rms}}} = \Theta(1) \quad \Rightarrow \quad \cos(\theta_{L-1}) = \Theta(L^{-1/2}).$$

This shows the result for the BFA at $g_{L-1}$ and the result holds as well for the BFA at $f_{L-1}$ up to hidden constants. $\square$

Interestingly, in view of Thm. (3.2), we can interpret the result of [Jelassi et al., 2023] as a computation on the spectral moments of a certain random matrix, as stated in the following lemma.

**Lemma A.1** (Spectrum of BFK and FBK in ReLU MLPs)**.** *For a ReLU MLP at random initialization satisfying (SP) and (BC), consider the BFK (at $g_v$ instead of $f_v$):*

$$K_v = \sum_{\ell \leq v} \eta_\ell \Big( \frac{\partial g_v}{\partial w_\ell} \Big) \Big( \frac{\partial g_v}{\partial w_\ell} \Big)^\top$$

*and $\theta_v$ the (non-oriented) angle between $\dot{g}_v$ and $z_\ell := (\partial \mathcal{L}/\partial g_v)^\top$. Then, in the notations of Thm. (3.2), it holds as hidden width diverges $\cos(\theta_v) = \big( M_1(K_v)/\sqrt{M_2(K_v)} \big) = \Theta(v^{-1/2})$.*

*Consider also, for a linear loss, the kernel $\tilde{K}_v$ such that $\dot{b}_v = \tilde{K}_v f_v$ (see Eq. (14)) and $\tilde{\theta}_v$ the (non-oriented) angle between $f_v$ and $-\dot{b}_v$. Then it holds, as hidden width diverges, $\cos(\tilde{\theta}_v) = \Theta\big( M_1(\tilde{K}_v)/\sqrt{M_2(\tilde{K}_v)} \big) = \Theta((L-v)^{-1/2})$.*

*Proof.* We have already seen in the proof of Prop.5.1 that $\cos(\theta_v) = \Theta(v^{-1/2})$. It thus remains to see that the assumptions of Thm. (3.2) are satisfied: the independence of $z_\ell$ follows from [Hanin and Nica, 2020, Prop. 2] and the Gaussianity of $z_\ell$ is direct since $z_\ell = W_{\ell+1}^\top b_{\ell+1}$ where $W_{\ell+1}$ is Gaussian and independent from $b_{\ell+1}$. Also, we have the more explicit expression

$$K_v = \sum_{\ell=1}^{v} \eta_\ell \|g_{\ell-1}\|_2^2 \left(\frac{\partial g_v}{\partial f_\ell}\right)\left(\frac{\partial g_v}{\partial f_\ell}\right)^\top$$

where $\frac{\partial g_v}{\partial f_\ell} = D_v W_v \ldots D_{\ell+1} W_{\ell+1} D_\ell$ and $D_i = \text{diag}(\phi(f_i))$ (by [Hanin and Nica, 2020, Prop. 2], these matrices can be taken as matrices with Bernoulli random variables on the diagonals, independent from everything else). Since under (SP) and (BC) we have that $\eta_\ell \|g_{\ell-1}\|_2^2$ is constant for $\ell \in [1 : L-1]$, it follows

$$K_v \propto \sum_{\ell=1}^{v} (D_v W_v \ldots D_{\ell+1} W_{\ell+1} D_\ell)(D_v W_v \ldots D_{\ell+1} W_{\ell+1} D_\ell)^\top.$$

For the second claim, we have (see Section 6) $\tilde{K}_v = \sum_{\ell=v+1}^{L} \eta_\ell \|b_\ell\|_2^2 \left(\frac{\partial g_{\ell-1}}{\partial f_v}\right)^\top \left(\frac{\partial g_{\ell-1}}{\partial f_v}\right)$. Under (SP) and (BC), we have $\eta_\ell \|b_\ell\|_2^2$ is constant for $\ell \in [2 : L]$, hence it follows

$$\tilde{K}_v \propto \sum_{\ell=v+1}^{L} (D_{\ell-1} W_{\ell-1} \ldots W_{v+1} D_v)^\top (D_{\ell-1} W_{\ell-1} \ldots W_{v+1} D_v).$$

By comparing the expressions for $K_v$ and $\tilde{K}_v$, we see that $\tilde{K}_v$ has the same distribution of nonzero eigenvalues as $K_{L-v}$ (potentially up to a global multiplicative factor) and the conclusion follows.  □

*Proof of Prop. 5.2.* The estimate for $\|f_\ell\|_{\text{rms}}$ is classical, see e.g. Li et al. [2021]. For the backward pass estimate, we rely on [Marion and Chizat, 2024, Lem. 3] (see also Zhang et al. [2022] for related results with the ReLU activation function), which implies that $\sigma_{\min}(\ell \to v) = \Theta(1)$ and $\sigma_{\max}(\ell \to v) = \Theta(1)$, where $\sigma_{\min}(\ell \to v)$ and $\sigma_{\max}(\ell \to v)$ are the smallest, respectively largest singular value of $\frac{\partial f_v}{\partial f_\ell}$. The estimate on $b_\ell = \left(\frac{\partial f_L}{\partial f_v}\right)^\top b_L$ directly follows.

For the BFA, we will apply the first bound of Thm. 3.2, namely $\cos(\theta_v) \geq \lambda_{\min}(K_v)/\lambda_{\max}(K_v)$ where $\lambda_{\min}(K_v)$ and $\lambda_{\max}(K_v)$ are the smallest, respectively largest, eigenvalues of $K_v$. In the forward pass (12), let us write $g_\ell = \phi(f_\ell)$ and $h_\ell = W_\ell g_{\ell-1}$. By direct computations, it holds (here $w_\ell$ is the vectorization of $W_\ell$):

$$K_v = \sum_{\ell=1}^{v} \eta_\ell \left(\frac{\partial f_v}{\partial w_\ell}\right)\left(\frac{\partial f_v}{\partial w_\ell}\right)^\top = \sum_{\ell=2}^{v} \eta_\ell \|g_{\ell-1}\|_2^2 \left(\frac{\partial f_v}{\partial h_\ell}\right)\left(\frac{\partial f_v}{\partial h_\ell}\right)^\top = \sum_{\ell=2}^{v} \eta_\ell \beta^2 \|g_{\ell-1}\|_2^2 \left(\frac{\partial f_v}{\partial f_\ell}\right)\left(\frac{\partial f_v}{\partial f_\ell}\right)^\top.$$

Using the inequalities

$$\lambda_{\min}(K) \geq \beta^2 \sum_{\ell=2}^{v} \eta_\ell \|g_{\ell-1}\|_2^2 \sigma_{\min}\left(\frac{\partial f_v}{\partial f_\ell}\right)^2, \qquad \lambda_{\max}(K) \leq \beta^2 \sum_{\ell=2}^{v} \eta_\ell \|g_{\ell-1}\|_2^2 \sigma_{\max}\left(\frac{\partial f_v}{\partial f_\ell}\right)^2$$

we deduce $\cos(\theta_v) \geq \lambda_{\min}(K_v)/\lambda_{\max}(K_v) = \Theta(1)$.  □

*Proof of Prop. 5.3.* In this proof, we say that a claim is found "by direct computation" when it can be directly deduced from the conclusion of Prop. 5.1. In particular, for the computation of scale invariant LRs, we use the fact that $\|\nabla_\ell \mathcal{L}\|_2 = \|b_\ell g_{\ell-1}^\top\|_F = \|b_\ell\|_2 \cdot \|g_{\ell-1}\|_2$. Also, by Prop. 5.1, under (SP) and (BC) it holds $\cos(\theta_{L-1}) = \Theta(L^{-1/2})$.

(i) One has that **MF+$\mu$P** satisfies (SP), (BC) by direct computation, and (FL) by Prop. 4.1. We have seen in the proof of Prop. 5.1 that $-\dot{\mathcal{L}} = \Theta(L^{-1/2})$, so (LD) does not hold.

(ii) For **NTK**, (SP), (LD) and (BC) can be checked by direct computation. For (FL), we have $\|b_{L-1}\|_{\text{rms}} = \Theta(1/\sqrt{m})$ so :

$$\|\dot{f}_{L-1}\|_{\text{rms}} = \Theta\left(\frac{1}{\cos(\theta_v) \cdot m \cdot \|b_{L-1}\|_{\text{rms}}}\right) = \Theta\left(\frac{1}{\cos(\theta_{L-1}) \cdot \sqrt{m}}\right).$$

But in the considered asymptotics $\sqrt{m} \cdot \cos(\theta_{L-1}) = \Theta(\sqrt{m/L}) \to \infty$ so $\|\dot{f}_{L-1}\|_{\mathrm{rms}} = o(1)$.

(iii) Properties (SP) specifies $\sigma_1$ and $\sigma_2 = \cdot = \sigma_{L-1}$, and Prop. 4.1 gives, with (FL), $\|b_{L-1}\|_{\mathrm{rms}} = \Theta(\frac{1}{\cos(\theta_{L-1})m}) = \Theta(\sqrt{L}/m)$ which imposes $\sigma_L = \sqrt{kL}/m$. Then the LR are given by (8). $\qquad\square$

*Proof of Prop. 5.4.* Properties (SP) specifies $\sigma_1$ and $\sigma_{\mathrm{hid}} = \sigma_2 = \cdot = \sigma_{L-1}$, and Prop. 4.1 gives, with (FL), $\|b_{L-1}\|_{\mathrm{rms}} = \Theta(\frac{1}{\cos(\theta_{L-1})m}) = \Theta(1/m)$ which requires $\sigma_L = \sqrt{k}/m$. Then the LR are characterized by (8). $\qquad\square$

*Proof of Prop. 6.1.* By the chain rule and Euler's identity for positively 1-homogeneous functions, it holds

$$b_v^\top f_v = \frac{\partial \mathcal{L}}{\partial f_v} f_v = \frac{\partial \mathcal{L}}{\partial f_L} \frac{\partial f_L}{\partial f_v} f_v = \frac{\partial \mathcal{L}}{\partial f_L} f_L.$$

Now, by differentiating in time both sides we get

$$\dot{b}_v^\top f_v + b_v^\top \dot{f}_v = f_L^\top \nabla^2 \mathrm{loss}[f_L] \dot{f}_L + \frac{\partial \mathcal{L}}{\partial f_L} \dot{f}_L = f_L^\top \nabla^2 \mathrm{loss}[f_L] \dot{f}_L + \dot{\mathcal{L}}.$$

We have $\dot{\mathcal{L}} = -\sum_{\ell=1}^{L} \eta_\ell \|\nabla_\ell \mathcal{L}\|_2^2$ and moreover, from the proof of Thm. 2.1, $-b_v^\top \dot{f}_v = \sum_{\ell \leq v} \eta_\ell \|\nabla_\ell \mathcal{L}\|_2^2$. So we deduce

$$-\dot{b}_v^\top f_v = -f_L^\top \nabla^2 \mathrm{loss}[f_L] \dot{f}_L + \sum_{\ell > v} \eta_\ell \|\nabla_\ell \mathcal{L}\|_2^2.$$

We conclude by writing $-\dot{b}_v^\top f_v = \|\dot{b}_v\|_2 \|f_v\|_2 \cos(\tilde{\theta}_v)$ and rearranging. $\qquad\square$

*Proof of Prop. 6.2.* By assumption at time $t = 0$, it holds $\tilde{\theta}(0) = \sigma \odot \theta(0)$ so let us prove the result by recursion. Assume that the claim is true at iteration $t$. Since $\prod \sigma_\ell = 1$, it holds $f_L(\theta(t)) = f_L(\tilde{\theta}(t))$. Moreover, since $\frac{\partial f_L}{\partial w_\ell}$ is 0-homogeneous in $w_\ell$ and separately 1-homogeneous in $(w_i)_{i \neq \ell}$. It follows

$$\nabla_\ell \mathcal{L}(\tilde{\theta}(t)) = \left(\frac{\partial f_L}{\partial w_\ell}[\tilde{\theta}(t)]\right)^\top \nabla \mathrm{loss}(f_L(\tilde{\theta}(t)))$$

$$= (\prod_{i \neq \ell} \sigma_i)\left(\frac{\partial f_L}{\partial w_\ell}[\theta(t)]\right)^\top \nabla \mathrm{loss}(f_L(\theta(t))) = \frac{1}{\sigma_\ell} \nabla_\ell \mathcal{L}(\theta(t)).$$

In particular, we deduce that the LRs are related by $\frac{\tilde{\eta}(t)}{\eta(t)} = \frac{\|\nabla_\ell \mathcal{L}(\theta(t))\|_2^2}{\|\nabla_\ell \mathcal{L}(\tilde{\theta}(t))\|_2^2} = \sigma_\ell^2$. It follows, for any $\ell \in [1:L]$,

$$\tilde{w}_\ell(t+1) = \tilde{w}_\ell(t) - \tilde{\eta}(t)\nabla_\ell \mathcal{L}(\tilde{\theta}(t)) = \sigma_\ell w_\ell(t) - \sigma_\ell^2 \eta(t)\frac{1}{\sigma_\ell}\nabla_\ell \mathcal{L}(\theta(t)) = \sigma_\ell w_\ell(t+1).$$

This proves $\tilde{\theta}(t+1) = \sigma \odot \theta(t+1)$ and the claim follows by recursion. $\qquad\square$

# B   Initializing with zero output weights

Let us mention an interesting degree of freedom for FSC in Table 1: up to adjusting the initial LR, it is possible to initialize the output layer with 0 while still satisfying FSC at the next step. If one initializes the output layer $W_L$ with 0 then all gradients are 0 at time 0 except that for $W_L$ which leads to the update (non-infinitesimal in this paragraph):

$$\delta W_L(0) = -\eta_L(0) \cdot b_L(0)g_{L-1}^\top(0).$$

The second forward pass is the same as the first one, with the only difference that

$$f_L(1) = -\eta_L(0)\|g_{L-1}(0)\|_2^2 b_L(0).$$

Assuming $b_L(0) = b_L(1)$ (linear loss) for simplicity, this leads to a second backward pass:

$$z_{L-1}(1) := \left(\frac{\partial \mathcal{L}}{\partial g_{L-1}}(1)\right)^\top = (-\eta_L(0)b_L(0)g_{L-1}(0)^\top)^\top b_L(0) = -\eta_L(0) \cdot \|b_L(0)\|_2^2 g_{L-1}(0).$$

For the second GD step to satisfy (FL), we just need to ensure

$$m\|z_{L-1}(1)\|_{\mathrm{rms}} = \Theta(\sqrt{L}) \qquad \Leftrightarrow \qquad \eta_L(0) = \Theta\left(\frac{\sqrt{L}}{m\|b_L(0)\|_2^2}\right).$$

This is the LR to be used at time 0, for the second step to satisfy (SP), (FL), (LD) and (BC).

*Proof of Thm. 6.3.* Prop. 6.2 shows that in fact only the product $\sigma_1 \cdot \sigma_h^{L-2} \cdot \sigma_L$ is a relevant degree of freedom of scale. We can thus fix (arbitrarily) $\sigma_1 = 1/\sqrt{d}$ and $\sigma_{\mathrm{hid}} = 2/\sqrt{m}$ so that (SP) is satisfied; we then have $\|b_v\|_2\|f_v\|_2 = \Theta(m\sigma_L)$ for $v \in [1:L-1]$. Desideratum (RFL) requires

$$\frac{\|\dot{f}_{L-1}\|_2}{\|f_{L-1}\|_2} = \frac{C_1 + C_h}{\cos(\theta_{L-1})\|b_{L-1}\|_2\|f_{L-1}\|_2} = \Theta(1) \qquad \Leftrightarrow \qquad \sigma_L \asymp (C_1 + C_h)\frac{\sqrt{L}}{m}.$$

using that $\cos(\theta_{L-1}) = \Theta(1/\sqrt{L})$. Moreover, (LD) requires $C_1 + C_h + C_L = \Theta(1)$. At this stage, the output scale $\sigma_L$ is not yet entirely determined since $C_1 + C_h = o(1)$ is not excluded. This is where (BS) comes into play. It requires in particular

$$\frac{\|\dot{b}_1\|_2}{\|b_1\|_2} = \frac{C_{\mathrm{hid}} + C_L}{\cos(\tilde{\theta}_1)\|b_1\|_2\|f_1\|_2} = O(1) \qquad \Leftrightarrow \qquad (C_{\mathrm{hid}} + C_L)\frac{\sqrt{L}}{m} = O(\sigma_L)$$

using that $\cos(\tilde{\theta}_1) = \Theta(1/\sqrt{L})$ by Lem. A.1. Combining both conditions for $\sigma_L$ imply, on the one hand, that $C_h + C_L = O(C_1 + C_{\mathrm{hid}})$ hence $C_1 + C_{\mathrm{hid}} = \Theta(1)$ and $\sigma_L = \Theta(\frac{\sqrt{L}}{m})$, which are equivalent to the constraints written in the theorem. Conversely, it is not difficult to see that these constraints lead to satisfying (RFL), (LD), (FS) and (BS). $\qquad\square$

## C   Characterization of reparameterization invariant LR

Consider a function $f : \prod_{\ell=1}^{L} \mathbb{R}^{p_\ell} \to \mathbb{R}$ admitting a (selection) derivative and, for a fixed scale vector $\alpha \in (\mathbb{R}_+^*)^L$ consider the function $g(y) = f(\alpha \cdot y)$ where $\alpha \cdot x$ denotes $(\alpha_1 x_1, \ldots, \alpha_L x_L)$. Consider one step of GD on the two functions, given for $\ell \in [1:L]$, by

$$x'_\ell = x_\ell - \eta_\ell \nabla_\ell f(x), \qquad\qquad y'_\ell = y_\ell - \eta_\ell \nabla_\ell g(y)$$

with identical starting points, that is $x_\ell = \alpha_\ell \cdot y_\ell$ for $\ell \in [1:L]$.

**Proposition C.1.** *Consider adaptive learning rates, which are of the form $\eta_\ell = \eta_\ell(\nabla f(x))$. Then $x' = \alpha \cdot y'$ for all $\alpha \in (\mathbb{R}_+^*)^L$ if and only if $\eta_\ell$ is $(-2)$-homogeneous in $\nabla_\ell f(x)$ and $0$-homogeneous in $\nabla_{\ell'} f(x)$ for $\ell \neq \ell'$.*

One such LR is precisely that suggested by Prop. 4.1: $\eta_\ell \propto \|\nabla_\ell f(x)\|_2^{-2}$.

*Proof.* For $\ell \in [1:L]$, it holds

$$\alpha_\ell y'_\ell = \alpha_\ell y_\ell - \alpha_\ell \eta_\ell(\nabla g(y))\nabla_\ell g(y) = x_\ell - \alpha_\ell^2 \eta_\ell(\alpha \cdot \nabla f(x))\nabla_\ell f(x).$$

Then $\alpha \cdot y' = x'$ for all $\alpha \in (\mathbb{R}_+^*)^L$ is equivalent to

$$\alpha_\ell^2 \eta_\ell(\alpha \cdot \nabla f(x)) = \eta_\ell(\nabla f(x)), \quad \forall \alpha \in (\mathbb{R}_+^*)^L$$

which is the claimed homogeneity property. $\qquad\square$

