# OpenReview forum: "The Feature Speed Formula: a flexible approach to scale hyper-parameters of deep neural networks"
_NeurIPS.cc/2024/Conference — NeurIPS 2024 poster_

### Official Review · Reviewer_PezB · 2024-07-11

**Soundness:** 3
**Presentation:** 3
**Contribution:** 2
**Rating:** 6
**Confidence:** 4

**Summary:**

The paper introduces the BFA - a novel quantity to predict and control feature learning in DNNs, as well as the feature speed formula which allows expressing the magnitude of feature updates after one GD step. The paper recovers key properties of known HP scalings, and also extends these results by introducing a new HP scaling for large depth ReLU MLPs.

**Strengths:**

1. The BFA and BFK are interesting objects to study and the geometrical picture that arises (mentioned in the introduction) gives a nice intuition.

2. The main results (Thms 2.1 and 3.2) are clearly stated and the proofs are straightforward.

3. The contributions are clearly stated and the relation to previous work distinguishes these contributions.

4. Earlier results are recovered here with a transparent derivation, but Ref [1] also provided quite an intuitive derivation, as you mentioned.





[1] https://arxiv.org/abs/2310.17813

**Weaknesses:**

Despite the strengths mentioned above, I did not give a higher score for the following reasons:

1. Novelty for HP scaling:
As far as I can see, the main takeaway regarding HP scaling is the extension of known results, such as muP, to the limit of large width-then-depth. While this is indeed new, this is a somewhat limited contribution.

2. Applicability of results:
While some of the results are rather general (like Thm 2.1), some other parts of the results seem to apply only under rather limited conditions, e.g. only a single input.

3. Experimental findings:
I found issues with some of the experimental findings: I did not find a mention of what is assumed about the data: is it synthetic, random, from some known benchmark etc. Also, by inspecting Fig 2b I was not convinced that the output sensitivity is bounded away from zero.

4. I feel that the paper could be made less technical and more readable by delegating some of the proofs to the Appendix and using the space for some visualizations.




typos:
- Fig1 caption 1st line: witdh -> width

**Questions:**

1. in line 194 - BC condition - is this a strict equality? or is there some tolerance?

2. In the Introduction you use the term "hierarchical features" - can you give a definition for that?

3. in the BFK definition (eq. 5) - is this for multiple inputs? the NTK is defined for $x, x'$.. Is $m_v$ here the product of batch-size with input dimension to the layer?

**Limitations:**

The authors adequately addressed the limitations of their results.

---

> ### Author Rebuttal · Authors · 2024-08-05
>
> We thank the reviewer for his/her detailed and encouraging review! Here is our answer to the main weaknesses raised by the reviewer:
> - in this paper, we focus on introducing a theoretical methodology and do not introduce new (useful) HP scalings indeed. In current work we are tackling other asymptotics via this approach (see answer to R#2 for an example).
> - the assumption of a single input is also made by all related works, but we do not believe that it is too problematic. The scalings are the same for a finite batch size as long as the NTK does not degenerate (eg on a ReLU ResNet -- a setting that we are developing in a work in progress). However, the situation would change and become more interesting for joint limits such as batch-size and width jointly going to $\infty$ (see also answer to R#2).
> - there was indeed a problem in the code of Fig.2b which will be fixed in the revision.
> - since our focus is on exposing a theoretical approach, we would like to keep the proofs in the main paper as much as possible. We will use the additional page to introduce illustrations and expand the discussions.
>
> Answers to questions:
> - the equality is indeed a typo, this should be a $\Theta(\cdot)$. Thank you!
> - we have removed the discussion on "hierarchical features" which we believe is not central to our goals and was too vague. By this term, we meant that each layer learns a representation of the input that relies on the learnt representation of the previous layer, and so on.
> - Yes the BFK in Eq 5 is for multiple inputs (in fact for essentially any architecture and tensor dimensions). In a vanilla MLP and Resnet, $m_v$ would indeed be the product of batch-size with the input dimension to the layer. It is this ability to cover any architecture and tensor shapes that we think make our approach a good starting point to derive HP scalings.

---

> > ### Comment · Reviewer_PezB · 2024-08-08
> >
> > thank you for your response.
> > I still did not find a "mention of what is assumed about the data: is it synthetic, random, from some known benchmark etc"
> > It would be helpful to get your response to that.

---

> > > ### Author Response · Authors · 2024-08-08
> > >
> > > We apologize for the oversight. These illustrations are computed with a single sample as an input (batch-size $1$) which is a random Gaussian vector in $\mathbb{R}^d$, so this is synthetic data. Since those models are rotationally invariant -- because the distribution of the input layer's weight is Gaussian -- we would get the same result for any input vector, be it synthetic, or from a dataset. We plan to release the (simple) code with the final version of the paper.

---

> > > > ### Comment · Reviewer_PezB · 2024-08-08
> > > >
> > > > thank you.
> > > > I will keep my score.
> > > > Good luck!

---

### Official Review · Reviewer_PDm8 · 2024-07-12

**Soundness:** 4
**Presentation:** 4
**Contribution:** 4
**Rating:** 8
**Confidence:** 4

**Summary:**

The paper presents a novel perspective on infinite width and depth feature learning networks. It introduces the backward-to-feature kernel (BFK) as a central quantity determining the evolution of the intermediate layer features. The paper shows that the movement of the hidden layer features can be exactly related to an angle $\theta_\ell$ between the backward pass and the feature velocity, and uses insights on the scaling of the cosine of this angle with width to recover essentially all known infinite width and depth feature learning limits, as well as a novel large depth MLP limit.

**Strengths:**

The paper studies an extremely important topic in deep learning theory. Given the technically challenging nature of the study of large width and depth limits, the paper is superbly well-written and accessible. Prior papers by Yang et al and Bordelon et al have done important work in developing the study of large width and depth limits, but their derivations are either very dense or otherwise rely on non-rigorous methods to derive the scalings. This paper manages to both rigorously motivate feature learning at infinite width and depth while simultaneously making the paper short and accessible. This is a major strength and no easy feat. I commend the authors on it.

Beyond this, there are several results of strong technical merit that will be of value for researchers studying infinite width and depth limits. The infinite depth MLP and scale invariant learning rate discussions are particularly interesting. The authors do a good job placing their work in context by presenting tables comparing their parameterization to others.

Ultimately, I believe that this paper is not only technically novel and sound, but is also a service to the community. I strongly recommend it for acceptance.

**Weaknesses:**

There are no technical weaknesses that I have found, and I have gone through the derivations in detail. My only comment is expository:
In equation 1, the definition of the forward pass $T_{\ell}(f_{\ell-1}, w_\ell)$ as well as its discussion in terms of selection derivatives is quite technical and may confuse readers from outside sub-communities in machine learning. I recommend stating more clearly that this includes a simple forward pass such as $W_{\ell} \cdot f_{\ell-1}$ and perhaps adding a footnote to make this first paragraph a bit more readable.

**Questions:**

As a simple clarification, I want to confirm that the ResNet scalings found precisely reproduce those predicted by Bordelon et al and Yang et al. Are there any additional ResNet scalings that have not been studied in prior work that this paper finds?

In the second paragraph of the conclusion section "it can only quantify feature speed for (S)DG (and does not apply to variants, a priori) and at “cut nodes” in the NN architecture, where all the signal goes through (in particular, it does not apply inside the blocks of a
290 ResNet)"

I assume you mean "(S)GD". Can you please elaborate a bit more on what you mean by cut nodes? Is this like a residual block with many layers? It will be interesting if you can derive a similar feature formula for more general differentiable circuits with branching.

**Limitations:**

Large width and depth limits may serve to determine the scaling laws for the next frontier of language and vision models, which may have major societal impact. However, the theoretical nature of the paper limits any major negative impacts.

---

> ### Author Rebuttal · Authors · 2024-08-05
>
> We thank the reviewer for his/her detailed and encouraging review.
> - yes the scalings for ResNets are precisely those predicted by Bordelon et al, and Yang et al, this is mentioned in the manuscript but we'll make this more visible (note that the first version of our work appeared in November 2023, the same month as Yang et al).
> - yes, we mean "SGD". By "cut nodes" we mean "cut nodes in the DAG computational graph where each node represents a tensor". This means that a ``cut node'' is any intermediate computations in the forward pass that is not skipped by any other computation, such as a skip connection in a Resnet. For instance, in a ResNet, the formula applies just after each residual connection, but not just before. We will clarify that (with a mathematical definition). We also think that it would be interesting and important to extend this approach to any node in any computational graph!
> - we agree with the comment that we should be a bit more concrete in the beginning of the paper, when introducing the forward pass and what this can represent. We will add examples of how to instantiate the formulas.

---

### Official Review · Reviewer_gz2f · 2024-07-14

**Soundness:** 3
**Presentation:** 2
**Contribution:** 1
**Rating:** 3
**Confidence:** 4

**Summary:**

The authors propose a technical strategy for deriving neural net parameterizations that relies on controlling the angle between the activation gradient and the feature update. The authors derive various theoretical results about this quantity, including a formula for computing it, and some analyses in the context of MLPs and ResNets. The authors claim to use this principle to derive new parameterizations, but crucially they never test them in a real learning problem.

**Strengths:**

- the authors propose an interesting notion and derive interesting analyses surrounding it
- the parts of the math I checked seem rigorous and sound
- the authors do a good job of connecting their work to related work
- the ideas are quite creative

**Weaknesses:**

I need to preface this review by saying that this feedback is intended to be constructive and to help you improve the paper. My current impression is that the paper is not ready for publication. I strongly encourage you to keep working in this direction, and I hope this feedback will be useful for that.

With that said, the main issues I see with the paper are:

### **No real experimental evaluation**

My understanding is that the main practical outcome of your work and theoretical analysis is a new parameterization for training neural networks. I feel that it is really important for you to test this parameterization to check that it is useful, or at least to see what its properties are in an actual training situation. It's so easy to come by free cloud compute (e.g. Google Colab) that I can't really see a reason for not doing this.

I don't feel that the experiments in Figures 1 and 2 are enough to convince me of the utility of your framework. Also I'm not sure how to reproduce these experiments. For example, what dataset did you use? What is the loss function?

As a side note, I'm also a bit doubtful that you can even train MLPs effectively beyond depth 20 or so. I read the Jelassi et al paper (https://arxiv.org/abs/2305.07810) and noticed they don't test their parameterization either. I may be wrong here, but I don't think you can hope for some engineer or experimentalist to pick up the paper and implement things for you. I think you have to be proactive here.

### **Doesn't go that far beyond existing ideas**
A lot of the paper focuses on dealing with analyzing or re-deriving existing parameterizations---e.g. muP or the 1/sqrt(L) depth scaling in ResNets. But this is not so interesting because it has already been done and there are already ways to analyze these things. What does your analysis offer that prior analyses do not? I also want to point out that concurrent works to this paper go beyond 1/sqrt(L) depth scaling rules. For example arxiv.org/abs/2405.15712 and arxiv.org/abs/2405.14813. And these papers actually experimentally test these deviations. Clearly these are concurrent works, but I just mention it to demonstrate that there is more out there.

### **Paper only seems to focus on batch size one**

In my opinion, doing things only at batch size one is a bit toy, and it would be better to directly analyze larger batch sizes.

**Questions:**

Please see the weaknesses section

**Limitations:**

Overall, I think it's hard to assess the limitations without having more thorough experimental evaluation. I would encourage you to work out how to streamline the mathematical exposition, and then to start testing these ideas.

I realize this feedback might be construed as being fairly negative, but I hope that it can help to improve the work.

---

> ### Author Rebuttal · Authors · 2024-08-05
>
> Thank you for your detailed review. We have replied to your first 2 criticisms in the main rebuttal. Concerning the limitation to batch-size 1: this is an assumption made by all related works on feature learning. However, note that the "feature speed formula" applies to any architecture -- and in particular to any batch size. With our approach, it is in fact possible to obtain the scalings in the *joint limit* of width and batch size -- which is more powerful than considering a large width limit with fixed batch size (and this is not accessible via other existing approaches, which start with the infinite width limit).
>
> For instance, a direct application of the feature speed formula in a two-layer MLP (one hidden layer) shows that to ensure feature learning, the scale of the output layer $\sigma_{L}$ should be scaled as
> $$
> \sigma_L \asymp \frac{1}{\cos(\theta_1)\cdot \text{width}} \asymp \frac{\sqrt{M_2}}{M_1}\frac{1}{\text{width}}.
> $$
> where $M_p$ are the spectral moments of the input covariance. (To derive the expression of $\cos(\theta)$, we use that in this case, the BKF is a linear operator that acts on a matrix $B\in \mathbb{R}^{\text{width}\times \text{batch}}$ as $K(B) = BX^\top X$, writing the linear case for clarity). This differs from the usual $\frac{1}{\text{width}}$ mean-field scaling (this new insight is not discussed in the manuscript to avoid overcrowding, but we mention it here to highlight the flexibility of our approach).

---

> ### Comment · Reviewer_gz2f · 2024-08-09
>
> I'll reply in full to your top level comment---thanks for writing that. Just two nits with this rebuttal:
>
> **"this is an assumption made by all related works on feature learning"** not true, for instance one of the papers I posted in my rebuttal (and it's non concurrent antecedents) does not make this assumption. It satisfies itself with proving bounds on feature learning that hold for any batch size.
>
> **"and this is not accessible via other existing approaches, which start with the infinite width limit"** my same comment applies again with the words "batch size" replaced by "width"

---

> > ### Author Response · Authors · 2024-08-09
> >
> > Thank you for engaging with my response.
> > - Indeed, I should have been more precise, I was talking about all works that provide proof of the scale of feature learning (by this, I mean both upper and lower bounds, of matching scale). The line of work of arxiv.org/abs/2405.15712 (using dynamical mean-field theory) is concerning with proving the existence of dynamical limits. The existence of well defined limits  can be interpreted as the existence of upper bound, but does not, however, imply lower bounds. The line of work of arxiv.org/abs/2405.14813 (on the "spectral criterion") is also, as far as I am aware, proposing a method that only gives upper bounds. Works that prove both upper and lower bounds (Yang et al., Jelassi et al. ) consider a single batch size, generally for convenience. But as I mentioned in my review, this is not out of reach with our approach.
> > - I am not sure I understood the sentence : "my same comment applies again with the words "batch size" replaced by "width" (since width is studied in our work)

---

> > > ### Comment · Reviewer_gz2f · 2024-08-09
> > >
> > > Thanks for engaging too. Okay I mainly agree with that characterization. The one remaining nitpick is that *"Works that prove both upper and lower bounds (Yang et al., Jelassi et al. ) consider a single batch size, generally for convenience"*. My understanding is that the fixed batch size assumption and infinite width assumption is critical to Tensor Programs analysis, since it essentially implies a low rank property of the gradients, giving you an a priori way to estimate gradient spectral norms. I can't speak for Jelassi et al, as I haven't been through that work carefully.
> > >
> > > And sorry, my cryptic comment was just supposed to say that the bounding approach can easily deal with finite width.

---

> > > > ### Author Response · Authors · 2024-08-09
> > > >
> > > > Thanks! Let us briefly comment on these points:
> > > >
> > > > - The fixed batch size assumption (not the *single* batch size one, which I was referring to) is indeed critical for the current Tensor Program framework (an algorithmic formulation of DMFT). They can handle random matrix (RM)/ random vector products but not RM/RM products, which output RM).
> > > >
> > > > - Regarding finite vs infinite width: our "feature speed formula" holds at any width as well. The "large size" asymptotics can then to be considered to obtain concentration of random quantities and sharp scalings (but, as we mention, it is not the only way one can use the formula).
> > > >
> > > > - You seem to be appreciative of the "spectral criterion" approach. So perhaps we should a be a bit more explicit: the theoretical results in https://arxiv.org/abs/2310.17813 (which was contemporary to our preprint) are strictly weaker than ours. By assuming bounded RMS to RMS spectral norms of the weight matrices (the spectral criterion), we can directly deduce upper bounds on the feature speed via our formula, in the fixed depth setting. However, this approach does not provides lower bounds (ie sharp scalings), nor gives depth scalings (and in general it is not clear when it can or cannot be trusted for general asymptotics). We are very appreciative of this line of work, but we feel that the strengths of our approach are not properly understood. On a side note, our formula also relies on some "spectral" information (via the first two spectral moments of the kernel which determines the angle $\theta$). This quantity is the fundamental spectral information that is needed to quantify feature learning, in any architecture, and any asymptotic.

---

> > > > > ### Comment · Reviewer_gz2f · 2024-08-09
> > > > >
> > > > > Thanks for your response.
> > > > >
> > > > > [Minor note: "at a single batch size" would usually be understood to mean "at a fixed batch size" and not "at a batch size of one". In other words, it would mean "at one value of the batch size" rather than "at a batch size of value one".]
> > > > >
> > > > > [Also a second note, that you may find interesting, but in actual transformer training (e.g. Llama-2-13b) we may have a batch size of 4 million tokens and a width of 5000, so actual transformer training is way outside the realm of applicability of muP]
> > > > >
> > > > > **"we feel that the strengths of our approach are not properly understood"** I take this seriously and I appreciate you've put a lot of work into your paper. I'll look over your paper again to check I haven't missed something.
> > > > >
> > > > > **"This quantity is the fundamental spectral information that is needed to quantify feature learning, in any architecture, and any asymptotic."** This is a huge claim. If you could demonstrate this in practice, everybody would want to read your paper. But as it stands you're not demonstrating it in **any** new situation.
> > > > >
> > > > > I won't reply further here as I want to move to writing a reply to your global response.

---

> > > > > > ### Author Response · Authors · 2024-08-10
> > > > > >
> > > > > > - Thanks, I understand our formulation "single batch size" is in fact confusing.
> > > > > > - I am aware that transformers are in a very different regime. This fact is one of the original motivations for our work ; to have a framework that does not relies on taking infinite width limit first (which is a strong limitation).
> > > > > > - The claim is perhaps huge, but it is indeed an equality, with a proof in the main text. The difficulty to make predictions in *new* settings are that (1) there is some theoretical difficulty in working out the various terms of the formula in general contexts (although it is not so difficult to get rules of thumb via, say, Marcenko-Pastur), and above all (2) it is not obvious what the desiderata should be, let's say when context length is very long. This is why we decided to make a paper where we focus on known results.

---

### Official Review · Reviewer_jmiK · 2024-07-17

**Soundness:** 1
**Presentation:** 3
**Contribution:** 2
**Rating:** 3
**Confidence:** 4

**Summary:**

This paper studies the feature learning speed of the layers of Neural Networks (NNs). Specifically, it proposes to measure it through the quantity *Backward-Feature Angle* (BFA), denoted by $\theta_l$ for a layer $l$. This quantity is directly related to the layer-wise decomposition of the Neural Tangent Kernel (NTK). In practice, the BFA is measured experimentally and several properties (feature learning, signal propagation, etc.) can be related to the BFA.

**Strengths:**

# Originality

This paper tackles an important problem: the relation between the optimal hyperparameters of a neural network and its architecture.

# Clarity

This paper is easy to read and the statements are clear.

**Weaknesses:**

# Originality

The BFA is closely related to the layer-wise decomposition of the NTK, which is already widely used in the NN optimization literature [1, 2, 3, 4]. Overall, the BFA does not contain any information that is not already available with previous objects.

# Significance

The benefits and the properties of the BFA are still unclear.

For instance, Section 5 proposes a new scaling of the hyperparameters, that is not clearly related to the BFA. Besides, the experimental validation of this new scaling is not provided.

# Quality

The contribution of this paper is unclear. The usefulness of the BFA, either theoretical or experimental, is still unclear, and the proposed hyperparameter scaling is not tested experimentally.

# EDIT: References

[1] Gradient descent provably optimizes over-parameterized neural networks (2018), Du et al.

[2] Gradient descent finds global minima of deep neural networks (2019), Du et al.

[3] A convergence theory for deep learning via over-parameterization (2019), Allen-Zhu et al.

[4] Stochastic Gradient Descent Optimizes Over-parameterized Deep ReLU Networks (2020), Zou et al.

**Questions:**

How does the "ours" hyperparameter scaling compare to the others (usual, muP or NTK)?

**Limitations:**

Lack of experimental validation.

---

> ### Author Rebuttal · Authors · 2024-08-05
>
> We have replied to this review in the main comment.
>
> Can you please specify what are the references [1,2,3,4] in your review? To the best of our knowledge our approach to derive HP scalings is new (the closest work being Jelassi et al), but we would appreciate precise pointers to related results.
>
> We are aware that the NTK and related objects have been heavily used in NN theory. This is expected since it is a fundamental characteristic of the training dynamics. However, we do not think any work has used these objects the way we do.

---

### Author Rebuttal · Authors · 2024-08-05

We thank the reviewers for their time and their comments and we appreciate their encouraging remarks. We also disagree with a few comments which, we believe, result from a misunderstanding of the content of our paper, the state of the theory on feature learning, and perhaps from a disagreement on the role of theory, in general.
- *lack of experimental validation*: this paper is theoretical and the usefulness of the hyper-parameter (HP) scalings that we discuss has been already demonstrated in prior works (which themselves followed from previous purely theoretical works on infinite width neural nets). The only new scaling that is introduced is the one for deep ReLU MLP but this just for the purpose to illustrate how to apply our theory in various contexts : as mentioned in the paper, deep ReLU MLPs cannot be trained (the NTK degenerates, a problem comes up when considering more than 1 sample) so there is no relevant "real data" experiment to do in this case.
- *lack of significance*: We sincerely believe that our approach is a significant advance in the theory of feature learning and HP scalings. The existing approaches to HP scalings are the following:

  (1) *write infinite-width limits dynamics first*, and see whether "features move" (eg Chizat et al, Yang et al, Bordelon et al, among many): this is the original approach to design HP scalings, but this leads to proofs which are technical and that help intuition in a limited way. For instance, in their breakthrough paper, Yang & Hu obtain a scaling for feature learning but only for specific architectures (namely finite depth tanh & gelu MLP) at the end of quite technical computations and use of a heavy random matrix machinery (see their appendix H.7.2/3, let us mention that this paper was obviously very influential to us). Note that the recent extensions of this approach to large depth limit are written only at a formal level, due to their high technicality. Besides, this approach is intrinsically limited to "infinite width first" limits and cannot deal with joint limits (e.g. with depth, batch-size or context length). All this facts make it crucial to dispose of alternative theoretical approaches to HP scalings.

  (2) *heuristics*: because of the difficulty of the above approach, other researchers typically rely on heuristics involving alignment/CLT/LLN etc. These heuristics can be useful as long as one lacks of rigorous and simple derivations (which we provide). For instance the "spectral condition" in (Yang, Simon, Bernstein) is a heuristic that recovers the good scalings in terms of "hidden width", but it fails to give the correct scalings for other asymptotics (such as depth or large batch-size).

In contrast, our approach is simple, rigorous and general. Since the "feature speed formula" is a non-asymptotic equality, it can be used as a starting point to obtain the good scalings in any asymptotic, provided each term is worked out : we have done this for width and depth in the current submission, and are currently extending it to other asymptotics (see also answer to R#2).

---

> ### Comment · Reviewer_gz2f · 2024-08-10
>
> Thank you to the authors for writing this. I was planning to reply to this global response with something about the philosophy of science and the role of theory (which are both topics I've thought about quite hard over the years) but in the interest of time, let me get to the point.
>
> **"as mentioned in the paper, deep ReLU MLPs cannot be trained"** I have gone over the paper three more times, and can nowhere find it mentioned that deep ReLU MLPs cannot be trained. On what line number do you say this? To me the paper gives the strong impression that you are proposing a new scaling for successively training deep ReLU MLPs. For example, you explicitly say that **"we introduce a new HP scaling (Table 1) that achieves both loss decay and feature learning"**
>
> In the discussions below, the authors state that they have found the fundamental quantity **"needed to quantify feature learning, in any architecture, and any asymptotic."** yet they do not validate this claim empirically in any setting and only produce scaling rules that are either already known or that the authors now admit will not work.
>
> ### **Suggestion to the authors for improving their work**
>
> This feedback is intended to be constructive. Of course, the authors may disagree with it and choose to ignore it, but here goes **I think you should take at least some responsibility for validating your ideas.** As I mentioned in my original rebuttal cloud compute is literally free at the moment (on Colab). All you need is an internet connection. You could run a simple test of your framework in under a day. You could then use this to sharpen your ideas, iterate on your ideas, and improve your science.
>
> I'm sorry this may not be the type of feedback you're hoping for. But in my view I think it's really important that somebody says this.

---

> > ### Author Response · Authors · 2024-08-10
> >
> > - I apologize for this: in fact, we forgot to include this comment about the degeneracy of MLPs in the submitted version; but it is in the arXiv version of the work, where we state: "let us mention that even though FSC fixes some degeneracies of deep MLPs, other problems arise when considering multiple inputs, such as degeneracy of the conjugate kernel and NTK [Hayou et al., 2019], which make ReLU MLPs a fundamentally flawed model at large depth." The case of MLPs is still an interesting case from a theoretical viewpoint because dynamical isometry does not hold, and it helps disentangling the pure effect of depth vs the effect of spectral degeneracy.
> > - Thm. 2.1 and 3.2 are *equalities* and Thm 2.1 is non-asymptotic besides the small learning-rate  (and Thm 3.2 requires the large width asymptotics, but in a robust way: it does not require width to be larger than any other quantity). So yes, we stand to the claim that anybody who attempts to quantify feature learning must be, implicitly or implicitly, trying to quantify the various terms in this formula.
> > - We sincerely appreciate all the time and energy that you have spent in this discussion. We agree that deep learning is fundamentally an empirical science and authors have the responsibility to validate their practical ideas. However, we disagree that it should be done for every single paper. There should also be room to develop intuitions into mature ideas, and to improve the understanding of known facts via alternative derivations, etc. Here, our paper's goal is solely to clarify the mechanism behind the scale of feature learning.

---

> > > ### Comment · Reviewer_gz2f · 2024-08-10
> > >
> > > Thanks for these clarifications.
> > >
> > > - I see now it's in the arXiv draft, but this feels quite an important omission from the submitted version
> > > - I see what you mean that Theorem 2.1 is an equality so holds quite generally. A good way to convince people this is a useful reformulation is to demonstrate how it can be used to do a new scaling calculation. As you mention below this may not be so straightforward.
> > >
> > > Pragmatically, I can't really engage with you further on this now (I still have two other papers in my batch to deal with). I've decided to maintain my score, and have the following suggestions:
> > >
> > > - If you want to commit to the pure theory route, which I'm not sure is the best thing, but I would suggest making another pass through the paper to clarify things as much as possible, include the omission about bad depth limits in MLPs. Another worthwhile contribution would be clarifying the connection to the spectral Yang et al paper. I know you pointed out these are concurrent works, but clarifying the connection is a strict benefit to the community, and you likely understand how to do this better than anyone else
> > > - My real suggestion is to do an actual practical training run of a neural network, and measure all your quantities
> > > - In terms of getting the paper past NeurIPS reviewers, this may be slightly jaded advice, but picking an architecture that people don't already know how to scale (e.g. Mamba, graph neural net, etc, etc) could be a good way to do this
> > >
> > > Good luck with your work

---

### Decision · Program_Chairs · 2024-09-25

**Decision:**

Accept (poster)

**Comment:**

The paper proposes the a quantity named BFA to predict and control feature learning in deep neural networks. Based on this, the authors derive a feature speed formula which allows expressing the magnitude of feature updates after one GD step. This paper receives diverging scores: two reviewers appreciate the strength of this paper in terms of soundness and sufficient technical contributions, while the other two reviewers highlight the weakness of the paper, mainly in not providing sufficient experimental verifications. After careful discussions amoung the reviewers, and between the AC and SAC, I am current leaning towards accepting this paper. The authors should include additional experimental results in the final version of the paper.